# Unsupervised Learning of Object Landmarks via Self-Training Correspondence

**Dimitrios Mallis**
University of Nottingham
dimitrios.mallis@nottingham.ac.uk

**Enrique Sanchez**
Samsung AI Center, Cambridge, UK
e.lozano@samsung.com

**Matt Bell**
University of Nottingham
matt.bell@nottingham.ac.uk

**Georgios Tzimiropoulos**
Queen Mary University of London, UK
Samsung AI Center, Cambridge, UK
g.tzimiropoulos@qmul.ac.uk

## Abstract

This paper addresses the problem of unsupervised discovery of object landmarks. We take a different path compared to existing works, based on 2 novel perspectives: (1) Self-training: starting from generic keypoints, we propose a self-training approach where the goal is to learn a detector that improves itself, becoming more and more tuned to object landmarks. (2) Correspondence: we identify correspondence as a key objective for unsupervised landmark discovery and propose an optimization scheme which alternates between recovering object landmark correspondence across different images via clustering and learning an object landmark descriptor without labels. Compared to previous works, our approach can learn landmarks that are more flexible in terms of capturing large changes in viewpoint. We show the favourable properties of our method on a variety of difficult datasets including LS3D, BBCPose and Human3.6M. Code is available at https://github.com/malldimi1/UnsupervisedLandmarks.

## 1 Introduction

Object parts, also known as keypoints or landmarks, convey information about the shape and spatial configuration of an object in 3D space, especially for deformable objects like the human face, body and hand. Prior research has focused on learning landmark detectors in a supervised manner for a few object categories only, for example facial [49, 5, 11] or human body keypoints [25, 2, 44], where thousands of annotated images with landmarks are available. As annotating landmarks for all objects is impractical, this paper focuses on learning landmark detectors in an unsupervised manner.

Unsupervised learning of object landmarks from a first glance seems an impossible task. A human annotator has understanding of the notion of objects and their parts, viewpoint invariance, occlusion and self-occlusion as well as examples of which landmarks to annotate in their disposal. Hence, it is completely unclear what machine learning task should be chosen for unsupervised landmark discovery with neural networks. Recent methods have focused on two principles/tasks: equivariance to 2D image transformations [40, 38, 37] and image generation [15, 51, 33]. These methods, despite presenting consistent results for various object categories, have their own limitations such as discovering landmarks with no clear semantic meaning (i.e. landmarks quite different to those annotated by humans)[1] and not generalizing well to large viewpoint changes (i.e. 3D rotations).

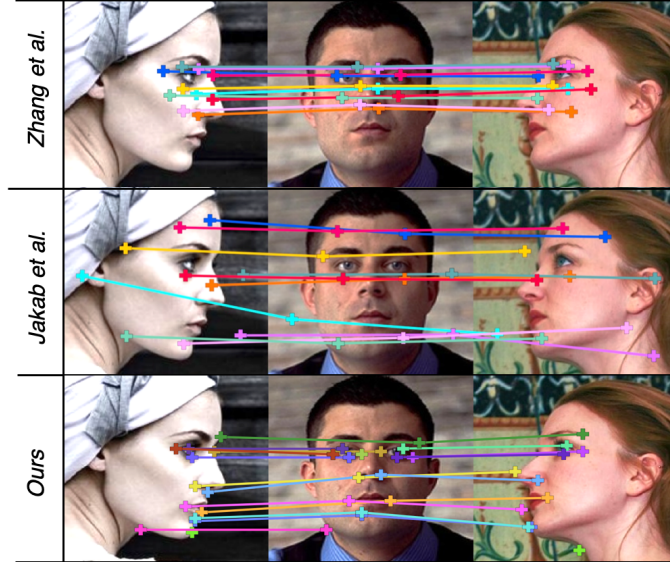

Figure 1: Comparison between the landmarks discovered by our approach and those by previous methods. Our approach provides landmarks that can better capture correspondence across large viewpoint changes. For example, there are visible landmarks detected in profile views that are matched with their corresponding points in the frontal view. Moreover, there are detected landmarks that are geometrically consistent in all 3 views. The method of [51] moves the point cloud altogether to fit the face region and hence loses correspondence (e.g. see red or orange point). Also, the method of [15] cannot cope well for such large changes in pose.

In this paper, we propose to take a different approach focusing on 2 novel perspectives:
**(1) Self-training:** for many object categories, manually annotated landmarks are mostly located on edges/corners of an object's surface, and hence they could be detected by a generic keypoint detector with good repeatability. Starting from generic, noisy keypoints, we propose a self-training approach where the goal is to train a landmark detector which learns to improve itself by discovering over the course of training more stable object landmarks and is becoming more tuned to them.

**(2) Correspondence via clustering:** one of the most important properties of a landmark detector is to achieve landmark correspondence across different images of the same category. We propose an optimization scheme which alternates between correspondence recovery via clustering and object landmark descriptor learning without labels. To our knowledge, this is the first time that recovering correspondence is applied to the problem of unsupervised discovery of object landmarks.

Compared to previous works, our approach can learn landmarks that are more flexible in terms of capturing changes in 3D viewpoint. See for example Fig. 1. We demonstrate some of the favourable properties of our method on a variety of difficult datasets including LS3D [4], BBCPose [8] and Human3.6M [14], notably without utilizing temporal information.

## 2 Related Work

**Unsupervised landmark discovery:** Only a few methods learn object landmarks without manual supervision. Several works approach this task by imposing equivariance to image transformations [40, 38, 37]. This constraint was also used in related tasks such as learning object frames [39], dense landmark detection [38], modelling of symmetric deformable objects [41] or learning latent 3D keypoints [37]. Somewhat related to our work is that of [38] which assumes that landmark detectors can be seen as local image descriptors. Building upon this assumption, the equivariance constraint is extended to make descriptors invariant to image exchange. Other approaches advocate the use of generative methods for image reconstruction. In [51, 24] object landmarks are discovered as an intermediate step of image auto-encoding. In [43, 15], the task is formulated by means of conditional

image generation that aims to reconstruct the input image from a deformed version of itself and the detected landmarks. In [33], this task is formulated from a domain adaptation perspective, producing more stable landmarks. A problem inherent to all these methods is that image generation is a proxy task unrelated to the task at hand and often yields landmarks which are uninformative. From a methodological perspective, we are the first to use (1) self-training, and (2) recovering correspondence via clustering for unsupervised learning of object landmarks. Also, none of the above methods has been shown to work well for large viewpoint changes which is one of the main results of our work.

**Self-training:** Self-training refers to a set of methods where a model's own predictions are used as pseudo-labels for model training. Usually, predictions are converted to hard labels and only retained when detected with high confidence [20, 35, 46, 29]. Multiple transformations of the same input sample can be also combined to form more accurate pseudo-labels [29]. Most self-training approaches focus on the task of image classification. More similar to our approach are methods for unsupervised segmentation [9, 17], foreground-background segmentation [12, 36] and salience object detection [48]. To our knowledge, there is no other framework based on self-training for unsupervised object landmark detection.

**Deep Clustering:** Our method can be linked with self-supervised approaches for representation learning through a clustering-based pretext task [47, 22, 45, 7, 27, 21]. Commonly, these approaches group the images into different clusters and a CNN is trained either to recognize samples belonging to the same cluster [21] or by using the cluster assignments as pseudo-labels [27, 7]. To our knowledge, learning object landmark correspondence via clustering in an unsupervised way is a novel perspective proposed for the first time in this work.

## 3  Method

### 3.1  Problem statement

Let $\mathcal{X} = \{\mathbf{x} \in \mathbb{R}^{W \times H \times 3}\}$ be a set of $N$ images corresponding to a specific object category (e.g. faces, human bodies etc.). After running a generic keypoint detector on $\mathcal{X}$, our training set $\mathcal{X}$ becomes $\{\mathbf{x}_j, \{\mathbf{p}_i^j\}_{i=1}^{N_j}\}$, where $\mathbf{p}_i^j \in \mathbb{R}^2$ is a keypoint and $N_j$ the number of detected keypoints in image $\mathbf{x}_j$. The original keypoints $\mathbf{p}^j$ for the $j$-th image are not ordered or in any correspondence with object landmarks. Also, multiple object landmarks will no be included in $\mathbf{p}^j$. Finally, some keypoints will be outliers corresponding to irrelevant background.

Using only $\mathcal{X}$, our goal is to train a neural network $\mathbf{\Psi} : \mathcal{X} \to \mathcal{Y}$, where $\mathcal{Y} \in \mathbb{R}^{H_o \times W_o \times K}$ is the space of output heatmaps representing confidence maps for each of the $K$ object landmarks we wish to discover. Note that the structure of $\mathcal{Y}$ implies that both order and landmark correspondence is recovered. Also note that K is the underlying number of "discoverable" object landmarks which our method aims to discover, not a hyperparameter. In practice, this number mostly depends on the dataset and the number of "good initial" landmarks detected by the generic detector.

We will break down our problem into 2 sub-problems/stages. The first stage aims to establish landmark correspondence, recover missing object landmarks and filter out irrelevant background keypoints. Then, the output of this stage is used to train a strong landmark detector. During this step, similar landmarks are grouped together.

### 3.2  Recovering correspondence

We will firstly learn a neural network $\mathbf{\Phi}$ with a shared backbone $\mathbf{\Phi}_b$ and two heads $\mathbf{\Phi}_{h,i}, i = 1, 2$ performing the following tasks:
**A detector head** $\mathbf{\Phi}_{h,1} : \mathcal{X} \to \mathcal{Z}$, where $\mathcal{Z} \in \mathbb{R}^{H_o \times W_o \times 1}$ is the space of single-channel confidence maps, which learns to detect all object landmarks with no order or correspondence, i.e. without distinguishing one from another (hence one output confidence map is used). The main purpose of $\mathbf{\Phi}_{h,1}$ is to recover the originally missed object landmarks.

The detector is trained to improve itself without labels through self-training. At every training round $t$, our method learns $\mathbf{\Phi}_{h,1}$ using the generated pseudo-ground truth landmarks at previous training rounds $t - 1$ and $t - 2$. The pseudo-ground truth landmarks are simply the model outputs after discarding those that are not close to a cluster centroid (see below). The detector is trained using an

MSE loss $L_d = \|H(\mathbf{x}_j) - \Phi_{h,1}(\mathbf{x}_j)\|^2$, where all landmarks $\{\mathbf{p}_i^j\}_{i=1}^{N_j}$ for image $\mathbf{x}_j$ are represented by a single heatmap $H$ with Gaussians placed at the corresponding landmark locations.

Our self-training approach confirms recent findings [1, 31] that show that over-parameterized neural networks tend to learn noiseless classes first, before overfitting to noisy labels in order to further reduce the training error. We observe such a pattern in learning object landmarks: a true landmark that commonly appears in the training set results in high detection confidence. Similarly, background locations that do not recurrently follow a specific pattern tend to be filtered out.

**A feature extractor head** $\Phi_{h,2} : \mathcal{X} \rightarrow \mathcal{D}$, where $\mathcal{D} \in \mathbb{R}^{H_o \times W_o \times d}$ for **recovering correspondence**. For each landmark $\mathbf{p}_i^j$, $\Phi_{h,2}$ computes a d-dimensional feature descriptor $\mathbf{f}_i^j$. This descriptor is used to cluster the landmarks into $M$ clusters each meant to represent an object landmark or different views/appearances of the same landmark. Our method computes the cluster centroids as well as cluster assignments using K-means. We also require that for a specific image no more than one keypoint can be assigned to the same cluster (i.e. object landmark). Hence, a modified K-means problem can be formulated:

$$\min_{C \in \mathbb{R}^{d \times M}} \frac{1}{N} \sum_{i=1}^{N} \sum_{j=1}^{N_j} \min_{\mathbf{y}_i^j \in \{0,1\}^M} \|\mathbf{f}_i^j - C\mathbf{y}_i^j\|_2^2 \quad \text{s.t.} \quad \mathbf{1}_M^T \mathbf{y}_i^j = 1 \text{ and } \|\sum_j \mathbf{y}_i^j\|_0 = N_j, \quad (1)$$

In practice, we solve the above problem (approximately) very fast by running the original K-means to find the cluster centroids and then using the Hungarian algorithm [19] to solve the linear assignment problem between the keypoints for an image and the cluster centroids, ensuring each keypoint is assigned only to a single centroid [2]. Furthermore, keypoints that are not close to any cluster centroid are filtered out.

We note that it is crucial to use $M \gg K$ ($K$ is the expected number of object landmarks). This enables our method to recover several different clusters per landmark which is necessary as viewpoint changes introduce large appearance changes. This differentiates our approach from prior works which do not account for large out-of-plane rotations. Merging the clusters corresponding to the same object landmark is handled at a later stage of the algorithm.

Given the keypoint-to-cluster assignments, we can recover landmark correspondence across images, and train $\Phi_{h,2}$ from paired images with known point correspondences. We want the features extracted at $\mathbf{p}_i^j$ to be close to those extracted at $\mathbf{p}_{i'}^{j'}$ if and only if $y_i^j = y_{i'}^{j'}$, and far otherwise, i.e. we want $\mathbf{f}_i^j \approx \mathbf{f}_{i'}^{j'} \iff y_i^j = y_{i'}^{j'}$. We formulate this objective in terms of a contrastive loss as:

$$L_c(i, i', j, j') = \mathbf{1}_{[y_i^j = y_{i'}^{j'}]} \|\mathbf{f}_i^j - \mathbf{f}_{i'}^{j'}\|^2 + \mathbf{1}_{[y_i^j \neq y_{i'}^{j'}]} \max(0, m - \|\mathbf{f}_i^j - \mathbf{f}_{i'}^{j'}\|^2), \quad (2)$$

where $\mathbf{1}_{[s]}$ is the indicator function, and $m$ is the margin. We form pairs from different images $j, j'$ as well as by letting $j'$ be a different augmentation of image $j$.

The overall training procedure for $\Phi_{h,1}$ and $\Phi_{h,2}$ is based on an alternating optimization and a self-training approach: clustering is performed at the end of each training round, and a new set of pseudo-ground truth locations is added to the training images.

### 3.3 Learning an object landmark detector

Given the pseudo-ground truth landmarks and their cluster assignments for all images in $\mathcal{X}$ provided by the method of Section 3.2, our final goal, in this section, is to train the landmark detector $\Psi$ (originally defined in Section 3.1). To this end, we simply train $\Psi$ to regress, for each training image $\mathbf{x}_j$, $M$ heatmaps $H_i, m = 1, \ldots, M$ each of which is a Gaussian placed at the pseudo-ground truth landmark location for that image. For a given image, the model is trained with an MSE loss over all output channels for which there is landmark-to-cluster assignment for that image: $L_d = \sum_m \|H(\mathbf{x}_m) - \Psi(\mathbf{x}_m)\|^2$. For clusters with no landmark assignments, we do not apply the MSE loss.

As mentioned in Section 3.2, many of the clusters capture the same landmark. Hence, during this step, we also perform progressive cluster merging. To this end, we take advantage of the structure of

$\mathbf{\Psi}$ (an hourglass network [25]) to set up a simple algorithm which takes into account both appearance and location related information: by construction $\mathbf{\Psi}$ can be decomposed into a shared backbone $\mathbf{\Psi}_b$, producing a feature tensor $\mathbf{F} \in \mathbb{R}^{H_o \times W_o \times d}$, and $M$ detectors $\mathbf{w}_m \in \mathbb{R}^{d \times 1}$ (implemented as $1 \times 1$ convolutions). If, say, two detectors fire at the same location $(k, l)$ (e.g. for a landmark that might have two close but different descriptors) then, they are both tuned to the same feature $\mathbf{F}_{k,l} \in \mathbb{R}^{d \times 1}$ at that location. Hence, detectors firing consistently at nearby locations (for the majority of training images) detect features of similar appearance and can be replaced by a single detector. We used this as the criterion for progressively merging different clusters, followed by training. At the end of this step, the number of detectors will be reduced from $M$ to $K$, where $K$ is the number of landmarks that is automatically discovered by our method.

### 3.4 Limitations

Thanks to SuperPoint initialization, our method appears to discover more semantically meaningful landmarks (especially for the human face) compared to previous methods. However, by no means, this guarantees a full semantic meaning for all discovered landmarks. Furthermore, although our approach provides landmarks that can better capture correspondence across large viewpoint changes, in many cases, the discovered landmarks are neither reliably detected nor able to provide full invariance to large 3D rotations. How to enforce consistency across different viewpoints is left for interesting future work.

Moreover, it is important to remark that the performance of our method greatly depends on the initial keypoint population (see also Section 4.1). On one hand, SuperPoint provides a strong initialization for our method. On the other, SuperPoint is prone to discovering salient points only, while other semantically meaningful landmarks lying on flat surfaces might be left unpicked. We note though, that most object landmark detectors mostly aim to detect salient points (e.g. mouth/eye corners, knee joints). Low texture points are difficult even for the supervised case. Finally, it is worth mentioning that although SuperPoint is trained in a self-supervised manner, it is bootstrapped by synthetically generated data that requires additional effort to produce.

## 4 Experiments

This Section presents experiments illustrating the results produced by our method and by recent state-of-the-art approaches, as well as ablation studies shedding light into some of the key properties of our method.

**Datasets:** We validate our approach on faces, human bodies and cat faces. For faces, we used CelebA [23], AFLW [18], and the challenging LS3D [4], consisting of large pose facial images. For CelebA, we excluded the subset of $1,000$ images (MAFL dataset [52]), which is used only to test our models. For AFLW we used the official train/test partitions, and for LS3D we followed the same protocol as [4] and used the 300W-LP partition [53] to train our models. For human bodies, we use BBCPose [8] and Human3.6M [14]. Although BBCPose and Human3.6M are video datasets, temporal information is not used. For results on Cat Heads [50] see supplementary material.

**Network architecture:** We use the Hourglass architecture of [25] with the residual block of [3] for both $\mathbf{\Psi}$ and $\mathbf{\Phi}$. The image resolution is set to $256 \times 256$. For network $\mathbf{\Phi}$, the localization head produces a single heatmap with resolution $64 \times 64$, and the descriptor head produces a volume of $64 \times 64 \times 256$, i.e. a volume with same spatial resolution containing the 256-d descriptors. The network $\mathbf{\Psi}$ produces a set of $K$ heatmaps, each $64 \times 64$, with the number of heatmaps being reduced throughout the training as described in Section 3.

**Training and implementation details:** Keypoints and descriptors are initially populated by Super-Point [10]. To increase the number of initial landmarks, we applied SuperPoint in 3 scales (1, 1.3 ,1.6). We applied K-means to SuperPoint descriptors to obtain the initial clusters and assignments. For K-means, we used the Faiss library [16]. We also applied an outlier removal step using the same Faiss library (this is not used later in the algorithm). Finally, bounding box information is used to discard the initial keypoints that are detected outside the object of interest (this is not used later in the algorithm).

For stage 1, for warm-up, we firstly trained both the detector and the descriptor for 20,000 iterations using as ground-truth the SuperPoint keypoints and their cluster assignments. Then, we trained

the model as detailed in Section 3.2, applying clustering and updating the pseudo-ground truth every 10,000 iterations. We set $M = 100$ and $M = 250$ clusters for facial and body landmarks, respectively. We found that no more than 300,000 iterations are necessary for the algorithm to converge for all datasets. Due to K-means sensitivity to centroid initialization, we do not initialize the centroids randomly but use the centroids of the previous training round. During landmark-to-cluster assignments (and since we have always a larger number of clusters than detected landmarks), to avoid poor assignments, we discard the assignment of a landmark to a cluster (and the landmark) when its distance to the cluster centroid is much larger compared to the average distance to that centroid.

For stage 2, we initialized the model from the weights of the model of stage 1, except for the weights of the last layer that are trained from scratch. We merged two clusters when $70\%$ of the number of points of the smaller cluster overlap with the points of the biggest cluster (the method does not seem to be so sensitive to this value). Overlap is defined as within 1 pixel distance at resolution $64 \times 64$.

To train the models, we used RMSprop [13], with learning rate equal to $5 \cdot 10^{-4}$, weight decay $10^{-5}$ and batch-size 16 for stage 1 and 64 for stage 2. All models were implemented in PyTorch [28].

**Evaluation:** Quantitative evaluation is often assessed by quantifying the degree of correlation between manually annotated landmarks and those detected by the proposed approach. This is accomplished by learning a simple regressor that maps the discovered landmarks to those manually annotated, using a variable number of images in the training set. The learned regressors are tested on the corresponding test partitions. However, our method does not detect a fixed number of landmarks per image, and hence training a regressor to predict the ground-truth annotations from a varying number of visible landmarks is not straightforward. In this paper, we opt for completing the missing values in the training set with the Singular Value Thresholding method for Matrix Completion [6] (leaving the detected points unchanged). For a given test image, we fill the missing values by the average landmark for that position, calculated from the training partition.

In addition, we follow [33], and complement this measure (herein referred to as **forward**) by measuring the performance of a reverted regressor, i.e. one that maps the manual annotations into the discovered landmarks. As found by [33], this measure, known as **backward**, helps identify unstable landmarks. Numerical evaluation is often measured by means of the Normalised Mean-squared Error (NME). However, we note that such a global metric does not help assess the performance of individual landmarks. To further examine the performance of both our and competing methods, we follow common evaluation practices in facial landmark localization and present CED (Cumulative Error Distribution) curves. Finally, for our ablation studies, to evaluate the quality of the descriptor head, we used the Normalized Mutual Information (NMI) [26] which is often used to assess clustering algorithms.

### 4.1 Ablation Study

**Feature representation:** We first evaluate the capacity of our method to learn distinctive features, corresponding to different landmark locations by computing the t-SNE [42] of the feature representations. Fig. 2a shows the t-SNE for the initial SuperPoint features, next to the features returned by our method after self-training. We observe that at the end of training, the descriptors are distinctive of the corresponding classes, making the correspondence recovery effective.

**Robustness to noise:** To study the impact of the quality of the initial keypoints, we replace the generic keypoints (provided by SuperPoint) by a mixture of (1) a varying number of ground-truth points randomly sampled from a set of 15 ground-truth landmarks, and (2) a set of noise points randomly sampled from the image domain. To evaluate the quality of the descriptor head, we measure the NMI between the assigned classes and the ground-truth labels. To evaluate the quality of the detector head, we report the F-measure between the detected points and the subset of the chosen ground-truth. Further implementation and evaluation details can be found in the Supplementary Material. We evaluate both the F-measure and the NMI with respect to the number of chosen clusters $M$, for a varying number of randomly chosen ground-truth points. The results shown in Fig. 2d illustrate that, when the initial keypoints include a $40\%$ of ground-truth locations, our method is capable of recovering the right correspondence. Note that Superpoint keypoints are of sufficient quality, too (see supplementary material).

**Impact of number of clusters:** In addition to the above, we report, in Fig. 2e, the NMI score for the detector w.r.t. a varying *number of clusters*. Even though keypoints are sampled from 15 groundtruth

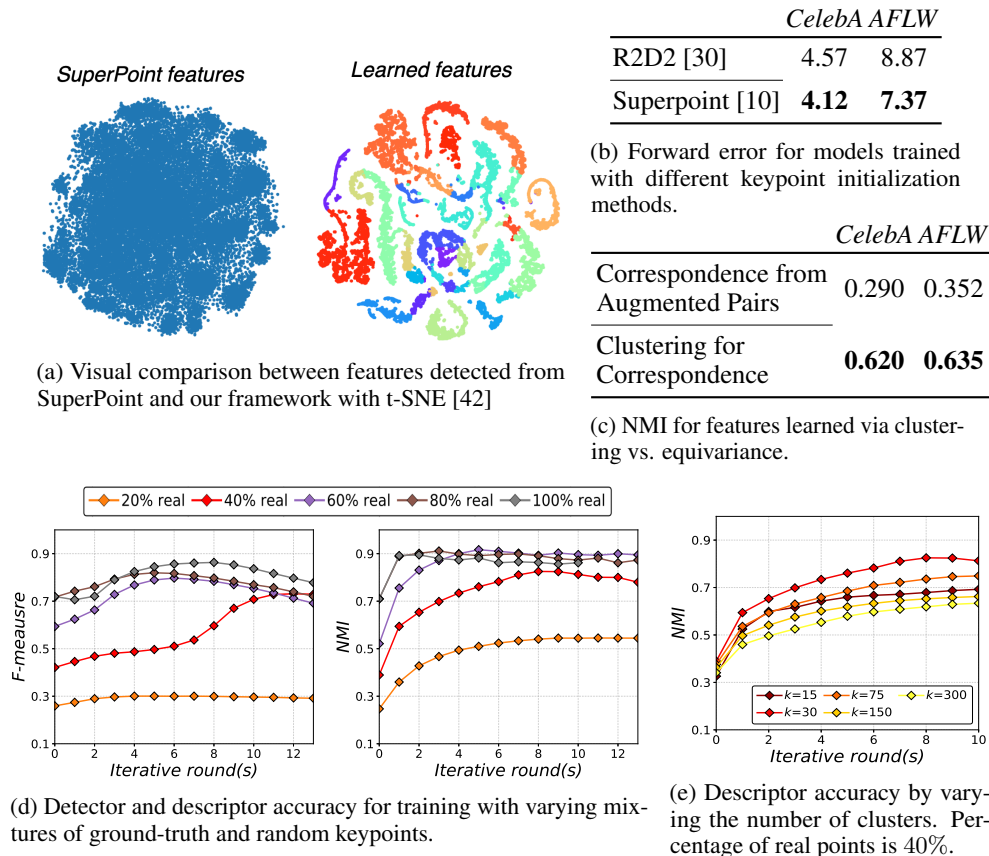

*SuperPoint features*     *Learned features*

(a) Visual comparison between features detected from SuperPoint and our framework with t-SNE [42]

|  | *CelebA* | *AFLW* |
|---|---|---|
| R2D2 [30] | 4.57 | 8.87 |
| Superpoint [10] | **4.12** | **7.37** |

(b) Forward error for models trained with different keypoint initialization methods.

|  | *CelebA* | *AFLW* |
|---|---|---|
| Correspondence from Augmented Pairs | 0.290 | 0.352 |
| Clustering for Correspondence | **0.620** | **0.635** |

(c) NMI for features learned via clustering vs. equivariance.

(d) Detector and descriptor accuracy for training with varying mixtures of ground-truth and random keypoints.

(e) Descriptor accuracy by varying the number of clusters. Percentage of real points is $40\%$.

Figure 2: Results for ablation study: Experiments depicted in sub-figures (d), (e) are performed on AFLW with synthetic keypoint initialization.

landmarks, we see that best performance is attained for $M = 30$. This over-segmentation of feature space is required for optimal clustering assignment, as it allows for multiple clusters that capture different appearance variations of the same landmark.

**Clustering vs. equivariance:** We evaluate the importance of recovering correspondence through clustering of local descriptors. To this end, we compare with a model trained without clustering at all but using image pairs produced by different affine transformations to compute the contrastive loss, i.e. the model was trained with equivariance. In Fig. 2c, we see that our approach outperforms the previous model by a large margin.

**Keypoint initialization:** We evaluate the influence of the initial generic keypoints in our proposed algorithm. To this end, we evaluate the performance of our method when initialized with the SuperPoint [10], as well as with the recent R2D2 [30], by means of to-tal forward error, evaluated on both CelebA and AFLW. Fig. 2b shows the results of the forward

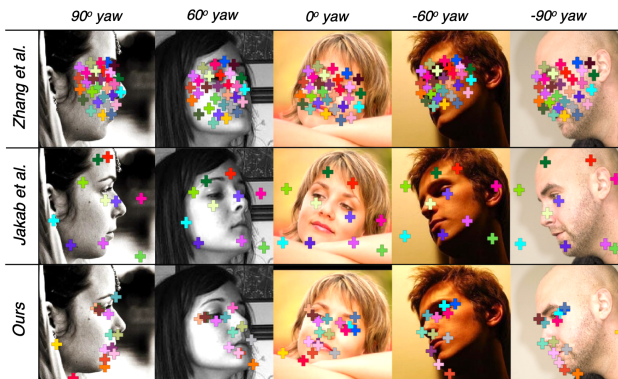

Figure 3: Comparison between landmarks discovered by our approach and those of [15, 51] on LS3D facial images across the whole spectrum of facial pose.

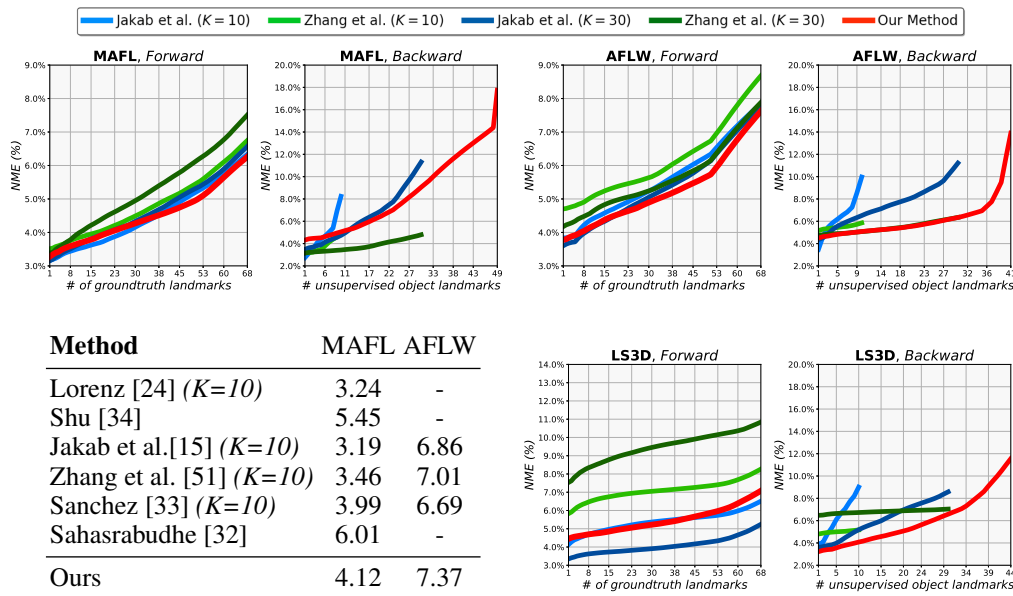

| Method | MAFL | AFLW |
|---|---|---|
| Lorenz [24] *(K=10)* | 3.24 | - |
| Shu [34] | 5.45 | - |
| Jakab et al.[15] *(K=10)* | 3.19 | 6.86 |
| Zhang et al. [51] *(K=10)* | 3.46 | 7.01 |
| Sanchez [33] *(K=10)* | 3.99 | 6.69 |
| Sahasrabudhe [32] | 6.01 | - |
| Ours | 4.12 | 7.37 |

Figure 4: Evaluation on facial datasets: Our method discovers 49 landmarks on CelebA, 41 on AFLW and 44 on LS3D. *(Table):* Comparison on MAFL and AFLW, in terms of forward error. The results of other methods are taken directly from the papers (for the case where all training images are used to train the regressor and the error is measured w.r.t. to 5 annotated points). *(Figures):* CED curves for forward and backward errors. A set of 300 training images is used to train the regressors. Error is measured w.r.t. the 68-landmark configuration typically used in face alignment.

evaluation of the generic keypoints provided by both methods. We observe that while both yield competitive results, SuperPoint is a better method for initialization.

## 5  Comparison with state-of-the art

Herein, we compare our method with [15, 51] trained to discover both 10 and 30 landmarks[3].

**Evaluation on facial datasets:** The bulk of our results on facial images is shown in Fig. 4. As discussed in [33], for a method to work well, both forward and backward errors should be small. From our results on all datasets, we can see that overall our method provides the best results in terms of meeting both requirements. Notably, our method delivers state-of-the-art results for the challenging LS3D dataset which is a dataset with frontal-to-profile pose variations.

In addition, a set of qualitative examples is shown in Fig. 3 for the challenging LS3D data. As opposed to our method, we observe that landmarks produced by [15, 51] are not stable under 3D rotations, and fail to capture large pose variations. More qualitative results for facial landmarks over different datasets, including the Cat Heads dataset, are included in the Supplementary Material.

**Evaluation on human pose datasets:** Performance of our method on the BBCPose and Human3.6M datasets is shown in Fig. 5. For both datasets, our approach demonstrates significantly better accuracy for both the forward and the backward errors. As it can be seen from the forward error in Human3.6M, all 3 methods experience a sharp increase of the error when more than 22 landmarks are considered. This is due to the fact that all methods did not capture the hands of the subject leading to very high error for the corresponding ground-truth points.

We also note that due to the large degree of pose variation for human bodies, a simple linear layer does not suffice to learn a strong mapping between unsupervised and supervised landmarks. Hence, the forward errors are very high for all methods. To address this, we follow [15] and measure the accuracy of unsupervised landmarks that are found to maximally correspond to the provided ground-truth

| | BBCPose Accuracy (%) | | | | | Human3.6M Accuracy (%) | | | | | | |
|---|---|---|---|---|---|---|---|---|---|---|---|---|
| | Head | Shldrs | Elbws | Hands | Avg | Head | Shldrs | Elbws | Waist | Knees | Legs | Avg |
| Zhang [51] | 95.36 | 32.36 | 15.02 | 28.02 | 42.69 | 20.90 | 53.05 | 50.95 | 43.70 | 85.60 | 1.95 | 42.69 |
| Jakab [15] | 48.29 | 18.55 | 19.10 | 32.11 | 26.83 | 0.50 | 52.15 | 32.35 | 26.05 | 3.70 | 24.6 | 23.22 |
| Ours | 83.65 | 62.93 | 63.13 | 35.00 | 61.18 | 95.1 | 59.15 | 58.60 | 67.59 | 69.35 | 73.30 | 70.35 |

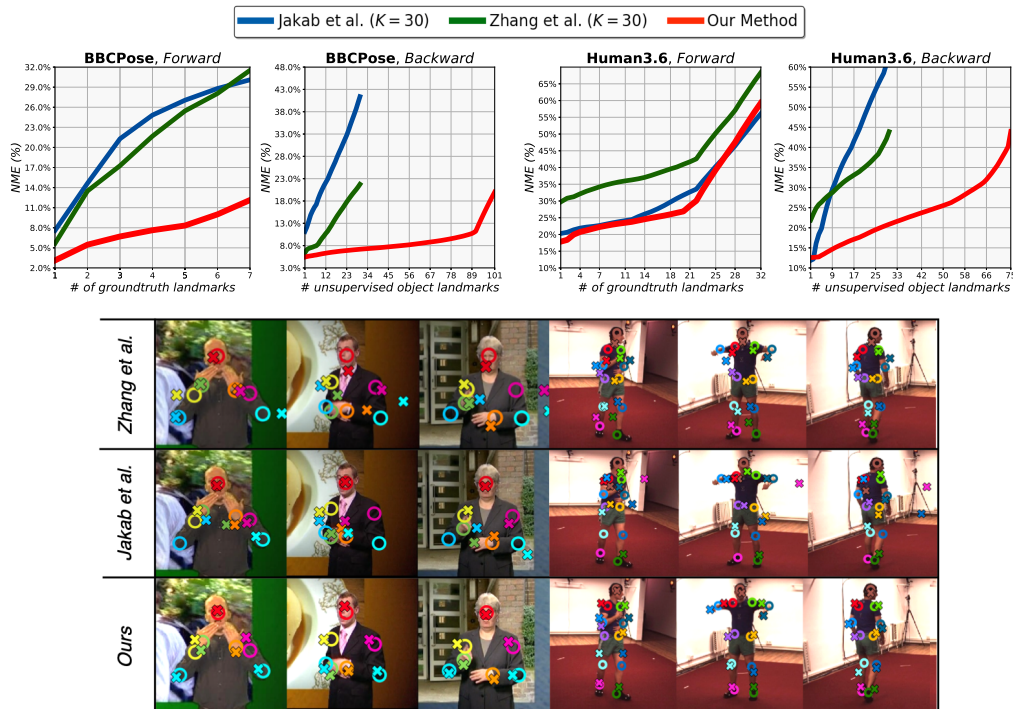

Figure 5: Evaluation on human pose datasets: Our method discovers 101 landmarks on BBCPose, and 75 on Human3.6M. *(Top)*: Accuracy of raw discovered landmarks that correspond maximally to each ground-truth point measured as %-age of points within $d = 12$px from the ground-truth [15]. *(Middle:)* CED curves for the forward and backward errors, computed for a regressor trained with 800 samples. *(Bottom)*: Visual demonstration of discovered landmarks (crosses) that maximally correspond to ground-truth keypoints (empty circles).

points (Table in top of Fig. 5). We can observe that our approach is able to discover clusters that robustly track all parts of the human body (except the hands for Human3.6M) and show much higher accuracy values compared to the other methods.

# 6 Conclusion

We presented a novel path for unsupervised discovery of object landmarks based on 2 ideas, namely self-training and recovering correspondence. The former helps our system improve by using its own predictions and constitutes a natural fit from training an object landmark detector starting from generic, noisy keypoints. The latter, although being a key property of object landmarks detectors, it has not been previously used for unsupervised object landmark discovery. Compared to previous works, our approach can learn view-based landmarks that are more flexible in terms of capturing changes in 3D viewpoint, providing superior results on a variety of difficult facial and human pose datasets.

## Broader Impact

This work presents a method for unsupervised discovery of object landmarks. This is a machine learning problem of extraordinary difficulty something that has been the main motivation for our work.

We would like to distance our approach from the problem of face recognition. Potentially, our method could be used to aid face recognition which has been occasionally criticized for serving harmful purposes. However, facial landmark localization is a mature technology already heavily used in industry, and large datasets with facial landmarks already exist. Moreover, all facial or human or animal related datasets used in our paper are standard datasets that have been heavily used by the research community.

## Funding acknowledgement

Dimitris Mallis' PhD studentship is funded by The Douglas Bomford Trust.

## Footnotes

[1]We note that our method cannot guarantee to discover semantic landmarks either, but tends to discover landmarks which are more semantically meaningful.

[2]Clustering followed by the Hungarian algorithm are performed at the end of each training round and are not part of the training the network.

[3]Where possible, we used pre-trained models provided, otherwise we re-trained these methods using the publicly available code.

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
