[Supplementary Material]

# Supplementary Material
# Unsupervised Learning of Object Landmarks via Self-Training Correspondence

**Dimitrios Mallis**
University of Nottingham
dimitrios.mallis@nottingham.ac.uk

**Enrique Sanchez**
Samsung AI Center, Cambridge, UK
e.lozano@samsung.com

**Matt Bell**
University of Nottingham
matt.bell@nottingham.ac.uk

**Georgios Tzimiropoulos**
Queen Mary University of London, UK
Samsung AI Center, Cambridge, UK
g.tzimiropoulos@qmul.ac.uk

This supplementary material includes some implementation and evaluation details that were not included in the paper due to space constraints, a detailed description of the datasets used to evaluate our approach, and further qualitative results in all datasets.

## 1 Further Implementation Details

**Experimental setup for ablation study:** We report a set of experiments on the ablation study investigating robustness to label noise and impact of number of clusters (Section 4 in the main submission). To perform these experiments, we populate each image with an initial set of points, mixed from a pool of ground-truth landmarks and a set of randomly chosen points. The initial number of points per image is randomly chosen to be between 12 and 24, so as to simulate the number of keypoints that are usually returned by SuperPoint. The ground-truth keypoint locations are sampled uniformly from a subset of 15 facial landmarks (3 on the nose and mouth, 2 on each eye and eyebrow and 1 on the jaw), while the remaining points are random 2D locations inside the facial bounding box. The ground-truth points are also randomly distorted within a 3 pixels radius.

To assess the quality of the pseudo-annotations produced by the first step of our approach (i.e. after the end of the step described in Section 3.2 in the main submission), the detector and descriptor head of our model are evaluated separately. Since the real ground-truth keypoints are sampled from a very specific subset of landmarks, an ideal detector would detect these 15 points in every image and filter out all noise. To evaluate how close our detector is from the ideal one we measure precision and recall combined with F-measure. F-measure for training with varying mixtures of ground-truth and random keypoints is reported in Fig. 2 (d) of the main submission.

To evaluate the descriptor part of the network we assess the information shared between the clustering assignments produced by our framework and the ground-truth landmark label for each detected keypoint. For a particular keypoint, the landmark label is that of the ground-truth to which it is maximally assigned, and the the clustering label is that assigned by the clustering of the corresponding descriptors. We measure the Normalized Mutual Information between the landmark and clustering assignments. This measure, that is independent of the number of clusters, can quantify the degree of which one assignment is predictable of the other. Normalized Mutual information of clustering assignments produced for the different settings examined in the ablation study are reported in Fig. 2 (c), (d), (e).

**Initialization quality:** To further validate that SuperPoint is a better choice of initialization than R2D2, we measure the recall provided by the initial keypoints w.r.t ground-truth points (3 on the nose

and mouth, 2 on each eye and eyebrow). The obtained recall was 0.42 and 0.29 for SuperPoint and R2D2, respectively showing that SuperPoint yields a stronger initialization. Moreover, Superpoint provides over 40% recall, that was shown in paragraph **Robustness to noise** of the ablation study in the main paper to be sufficient for our approach to recover the correct correspondence through self-training.

## 2  Datasets

Our approach was evaluated over the following datasets.

**CelebA-MAFL:** CelebA [6] dataset of about 200K facial images annotated for 5 facial landmarks. Following standard practice we evaluate our method on the MAFL subset, which is excluded from the training split. Calculation of the forward and backward error curves is performed w.r.t. 68 standard facial landamrks that are recovered using the highly accurate method of [1].

**AFLW:** AFLW [5] contains $10,112$ training images and $2,991$ test images annotated for 21 landmarks annotated based on visibility. Similar to CelebA, calculation of the forward and backward error curves is performed w.r.t. 68 standard facial landamrks that are recovered using [1].

**LS3D:** LS3D [1] is a dataset of large pose facial images constructed by annotating the images from 300W-LP [11], AFLW [5], 300VW [8], 300W [7] and FDDB [4] in a consistent manner with 68 points using the automatic method of [1]. Note that LS3D dataset is annotated with 3D points. Evaluation is performed on the LS3D-W Balanced test set including an equal number of images for yaw angles of $[0^o - 30^o], [30^o - 60^o], [60^o - 90^o]$.

**Cat heads:** Cat heads [9] contains images of cats annotated for 9 facial landmarks, 2 for the eyes, 1 for the mouth and 3 for each ear. The dataset is divided into 7 disjoint folders. We used 6 for training ($\sim 8750$ images) and 1 for testing ($\sim 1250$ images).

**BBCPose.** BBCPose [2] is a dataset of 20 sign language videos annotated with 7 human pose landmarks (head, wrists, elbows, and shoulders). Evaluation is performed on the standard test set of 1000 images.

**Human3.6M.** Human3.6M [3] is an activity dataset with a constant background containing videos of actors in multiple poses under different viewpoints. Similarly to [10] we use all 7 subjects of the training set: 6 subjects were used to train our model and 1 was used for testing.

## 3  Qualitative results

For qualitative results on various datasets please see the figures in the following pages.

Figure 1: Qualitative results on MAFL

Figure 2: Qualitative results on AFLW

Figure 3: Qualitative results on LS3D

Figure 4: Qualitative results on Cat Heads

Figure 5: Qualitative results on BBCPose

Figure 6: Qualitative results on Human3.6M