[Reviews · NeurIPS 2020]

Review 1

Summary and Contributions: This paper is on unsupervised landmark discovery, and the big improvement on prior work is how they can handle large variations in pose, such as profile faces. They make a connection between semantic landmarks and generic keypoint detectors (in the classical computer vision sense of "Good features to track" - edges, corners) and pose the problem as one of refining a noisy collection of keypoints to find those which can be used as a basis for stable landmarks. The method involves iterative self-training in a similar vein to DeepCluster. A one-channel output heatmap is trained to give maxima at the initial generic keypoint locations. They observe that strong responses occur at the most stable, non-background landmarks and so are suitable to be used as labels for further training. A separate 256d descriptor head is used to cluster in descriptor space, allowing to for example say that the left eye is the same semantic thing in two different images. The detections and cluster assignments can then be used as labels to train a landmark detector, with some further cluster merging needed to group detectors that fire in the same locations and become robust across large view shifts. Compelling results are given on standard benchmarks, and the strength across large pose differences is shown on the more challenging LS3D.

Strengths: Overall it is a good paper and brings something new to the area of unsupervised landmark discovery. The claim that the method finds more robust landmarks when it comes to large viewpoint changes is clearly demonstrated in qualitative results such as Fig1 and Fig4, where the weaknesses of previous methods are very obvious. This is backed up by quantitative results on LS3D in particular. On Human pose they show good quantitative results. The ideas presented are interesting and useful, and there is a significant amount of novelty compared to how this problem has been solved in prior work. Using generic keypoints as a starting point makes a lot of sense, and the mechanism of using a CNN and self-training to find which keypoints are most reliable is quite interesting. The proposal to use clustering to obtain inter-instance correspondences appears to be a novel approach to unsupervised landmark learning, and given that the overall iterative scheme seems to work well I think it will be relevant to others in the community and could inspire future work on extracting object structure beyond just landmarks. Another strength of this method is that it allows a landmark not to fire if it is not present in the image, whereas most other methods assume that all the landmarks are visible in every image and will do something weird if they are occluded, like have spurious detections in random locations or squash them into the remaining space on the object. So this is a very welcome solution to that problem.

Weaknesses: Some discussion on the failure modes would be welcome. In particular it would seem that by using generic keypoints as an initialisation, landmarks would not be found in areas of low texture that are nonetheless semantically interesting. Think a centre-of-forehead landmark for example, or the chin contour annotated in 68pt landmark datasets. Some mis-detections also seem to occur, such as the blue right-eyebrow landmark appearing on the left eyebrow in Fig4 col5 and the pink left-eyebrow landmark not appearing (where I use left/right wrt face not viewer). Not discussed what design decisions about the method make it work well for big out-of-plane rotations. Is it the grounding in generic keypoints that pins landmarks to real distinctive features that move in 3d space or the clustering and subsequent cluster merging across views? Some analysis here would be nice. It is outperformed by other methods on MAFL/AFLW - although these datasets don't tackle large variations in pose where this method really shines. Quantitative results on 300W would be nice, since this has also been used as a benchmark for unsupervised landmarks and is more challenging (with 68 landmarks). For the LS3D Forward curve - I think the improvement is clear in the lower left, but disappears as it reaches 68 landmarks. So perhaps due to the limitations mentioned above it is really good at the central face region where there are good anchors for generic keypoints but bad at the outer face/chin countour. It would be good to see the per-landmark errors to know if that is indeed the case.

Correctness: I have a doubt about the correctness of the evaluation: Is any labelled data used to train the generic keypoint detector? I see that in the SuperPoint paper they pretrained on a synthetic dataset of millions of shapes. If so the comparison to purely unsupervised methods is not really fair, and I would recommend showing results with a purely unsupervised/self-supervised pipeline (cf Pathak et al CVPR17 "In order to have a purely unsupervised method, we replace the trained edge detector in NLC with unsupervised superpixels."). It would be good to clear this up and get a good assessment of what can be learned "from pixels alone".

Clarity: The paper is well written and easy to follow - no problems here. Spelling error: "Correspondance"

Relation to Prior Work: As far as I can tell all relevant prior work is mentioned, and it is made clear how this work differs and tackles limitations of previous work.

Reproducibility: Yes

Additional Feedback: Fig3 CED: I didn't think MAFL had 68pt annotations, is that right? Actually I can answer my own question, as per sup mat these were obtained using a supervised method - maybe worth saying in the main text since it was a bit puzzling. ----- Post rebuttal ----- My recommendation is acceptance, however it is important that the authors address the issues raised by the other reviewers, in particular on clarity. It is also important that the authors make very clear the effect of the SuperPoint initialisation and the level of supervision this has. In the rebuttal they claim "Superpoint is trained in a self-supervised manner. So we still believe it qualifies for unsupervised" however this is not completely true as Superpoint uses synthetic data in training (see fig4 of Superpoint). As the authors point out in the rebuttal, other work has used pre-trained networks such as perceptual loss, so it's not a huge problem it just needs to be made clear so methods can be judged fairly.


Review 2

Summary and Contributions: This paper presents an unsupervised method for keypoint detection in images. The main idea is to combine keypoint detection and description with correspondence. The latter then guides to detect more distinctive keypoints even in the absence of annotated ground truth. The method is evaluated on a range of different datasets, primarily consisting of human faces, human bodies and cat faces and shown to perform either better or on par with supervised detection methods.

Strengths: I think the main strength is the unsupervised nature of the method. The idea of combining keypoint detection with correspondence is also, to the best of my knowledge, novel and can lead to follow-up works.

Weaknesses: Some of the quantitative results are not particularly impressive such as the comparison on facial datasets, shown in Figure 3 of the paper. Furthermore, another major weakness is that the method needs to be initialized with keypoints detected by a generic detection method (SuperPoint is used in the paper). Therefore, any significant failure in that step will potentially negatively impact the entire pipeline.

Correctness: Seems good, although I did not verify all the details.

Clarity: The paper is generally clearly written. Some details were somewhat unclear to me, however, including the choice of K (expected number of object landmarks) which seems to be a hyper-parameter. I also did not quite understand how the authors used the Hungarian algorithm in their training procedure. That algorithm is not differentiable, as far as I understand. Therefore, it's not clear to me how the pipeline which includes this step can be trained end-to-end. A discussion about this would be very useful.

Relation to Prior Work: I'm not an expert in this area, but the main new insight seems to be the use of correspondence inside an unsupervised (or perhaps more aptly, weakly supervised, since it requires a pre-trained keypoint detector as initialization) method. I think that's an interesting idea that can lead to follow-up work.

Reproducibility: Yes

Additional Feedback:


Review 3

Summary and Contributions: The paper addresses an important problem of unsupervised landmark discovery. It distils objects landmarks from generic keypoints that are initialized using an existing pre-trained method - Superpoint. These generic keypoints are trained to generalize under large viewpoint changes and that helps the proposed method to learn more stable object landmarks than other unsupervised methods. This is a great contribution. Unfortunately, the paper lacks clarity and the experimental section is weak.

Strengths: It is a very neat idea to try to distil object landmarks from generic keypoints and I really like that it generalizes well under large viewpoint changes when compared to other methods. Figure 1 is good for illustrating the strengths of the method.

Weaknesses: While it has its strengths, the method uses a strong initialization by building on top of a pre-trained interest-point detector - Superpoint. This is in contrast with other unsupervised methods for object landmark discovery that learn to discover object landmarks from scratch. This is not a flaw in particular but it should be sufficiently highlighted when comparing to previous art. The writing in the method section lacks clarity. The experimental section should be improved by comparing to other state-of-the-art methods for all the metrics such as [22] . The method shows very good generalization under large viewpoint changes but this is demonstrated only on one dataset. It would nice to see that evaluated on other datasets of 3D objects. In summary, this could have been a very strong paper if more time was invested in the experiments and the method description was clearer. In the current form, the paper is only half-baked. Please see the detailed comments below.

Correctness: The experimental section is weak. It ommits comparisons to other state-of-the-art methods. The strength of this method (generalization under large viewpoint changes) is demonstrated only on one dataset LS3D. See the detailed feedback bellow.

Clarity: The papers is not clear enough. Definitions of variables are often missing. Inconsistent language (keypoints/landmarks/object landmarks) makes the paper confusing. Illustration of the method would help understanding.

Relation to Prior Work: The fact that this method uses strong keypoint initialization while other comparable methods learn them from scratch should be more discussed. Other comparable methods are dissuced in the related work but are not compared to.

Reproducibility: No

Additional Feedback: Detailed feedback: 1. Authors state that "[other] methods, despite presenting consistent results for various object categories, have also their own limitations such as discovering landmarks with no clear semantic meaning." This claim is rather strong since the proposed method also does not guarantee any clear semantic meaning for object landmarks discovered by their method. 2. Comparison with [22] is missing for many experiments. That work actually reports better accuracy on BBCPose than the proposed method and hence should be also included. 3. A schematic illustration of the method would improve clarity. 4. Incosistent use of "keypoint", "landmark" and "object landmark" words. Since the paper is trying to distinguish between "keypoints/landmarks" and "object landmarks" it would be helpful to have a clear definition and use them consistently. For example, in the introduction, the three words are used interchangeably but then in the section 3 "keypoints and landmarks" refer to very different entities than "object landmarks". That makes the paper confusing. 5. Variables in eq. 1 are not properly defined. The reader has to infer what they are supposed to represent. 6. p_{i'}^{j'} is not defined 7. Since the method heavily relies on a strong keypoint initialization when compared to other methods that do not require this, it would be useful to also see ablations with weaker methods like SIFT. 8. Authors show that initialization with more recent method R2D2 leads to worse results than SuperPoint. As this is surprising, an they provide any insight why they think this happens? 9. Figure 5 bottom second row Human3.6M experiments looks suspicious. If [13] is trained with a static background then there should not be any discovered landmarks on the background. 10. The strength of this method - generalization under large viewpoint changes should be demonstrated on other datasets like 3D objects. 11. Since the method seems to learn stable object landmarks under large view-point changes, comparison with unsupervised 3D landmark methods like [Suwajanakorn, Supasorn, et al. "Discovery of latent 3d keypoints via end-to-end geometric reasoning." NeurIPS. 2018] would be really interesting. Typos and style: 23: "his/her" should be replaced with"their" pronoun ------------------- Post rebuttal ------------------- Given that the authors addressed some of my concerns in the rebuttal, I am rising my rating. However, authors have to address rest of the reviewers' concerns in the final version. Most importantly, be clear about the level of supervision their method requires, strong reliance on Superpoint method that is pre-trained on synthetic data, and be fair when comparing to previous methods that learn to discover landmarks "from scratch" while usually requiring less supervision input.


Review 4

Summary and Contributions: This work aims at unsupervised discovery of landmarks in image collections of a specific object category, e.g. faces and humans. They achieve this through a novel approach of: (1) Boostrapping using self-learning/pseudo-labels generated from their own model, starting with a generic keypoint/interest point detector, while simultaneously clustering these detections based on their image features. (2) Using the discovered cluster assignments to establish correspondence across different instances/images. They present quantitative evaluation on human face datasets --- MAFL, AFLW, LS3D, and human-pose -- BBCPose and Human3.6M, where they are comparable to sota on the standard "forward" (regressor from discovered -> gt points), and substantially better on the "backward" metric (regressor from gt -> discovered), highlighting successful rejection of "noisy" keypoints.

Strengths: The key strength of this work is that it proposes/applies a fresh approach to discovering landmarks --- and that is one of refinement (or boostrapping/self-learning) and feature clustering, which although has been applied to general feature learning for classification, is under-explored for unsupervised landmark discovery. Previous works rely on equivariance to synthetic transformations or expensive image-generation. This work instead relies on self-training/ training on pseudo-labels bootstrapped from a generic keypoint detection (which is in turn trained using equivaraince [SuperPoint]). Empirically, the method discovers keypoints which are more aligned with the ground-truth ones (as highlighted by "backward" keypoint metric). However, the "forward" accuracy is comparable (both slightly better/worse depending on the dataset) to sota methods. The topic is highly relevant, but the impact might be limited due to the complexity of the proposed approach.

Weaknesses: 1. A key limitation is that the proposed method is much more complex than the sota methods for unsupervised keypoint discovery. Further, the gains in performance are not commensurate with the additional complexity, which might lead to limited impact of this work. 2. Another limitation is that on "standard" inter-occular normalized error metrics used in related works for unsupervised facial landmark detection, the method seems to underperform on both MAFL and AFLW as compared to the method of Jakab et al. [13] and Zhang et al. [48]. 3. While the method does improve on sota on the "backward" error metric (measured by training a regressor from ground-truth to unsupervised discovered keypoints), comparisons against the method which they cited for this metric (Sanchez et al. [31]) are missing. 4. They use much higher resolution (256x256) input images than previous work of Jakab et al. and Zhang et al. (~100-128 pixels squared). Hence, the accuracies may not be directly comparable. 5. It is not clear if the clustering in the 2nd phase of training (section 3.3) is required at all if a linear regressor is learnt on top of the discovered keypoints for evaluation anyway. A clear ablation on this should be presented. 6. SuperPoint itself is trained using equivariance. It would help if a compaison against more traditional interest point detectors (e.g. SIFT) is presented.

Correctness: The empricial methodology is correct, however, the presentation is missing many details which makes it very difficult to follow. Please refer to "clarity" below. Sota unsuperivsed landmark discovery methods [36, 31] are cited but are missing in sota comparison tables/plots. Please include these for completeness. For a missing reference, see "prior work" below.

Clarity: Although the description of the method itself is satisfactorily detailed, deeper details are missing, which would affect reproducibility: 1. The text in section 3.3 (lines 142-152) describing the clustering procedure of keypoints is rather unclear: (a) How does Hourglass architecture help in the construction of appearance and location heads? (b) What do `w_m` correspond to? What are their inputs/outputs? (c) Why would two detectors fire at the same location? (d) What is `F`? (e) What is the stopping criterion for clustering? What are the "discovered" values of `K` -- how do they correspond to the ground-truth for various datasets? 2. How is the "detector head" \Phi_{h,1} bootstrapped? How are is "pseudo-ground truth" generated---is some kind of thresholding performed? Do you also use the output of the "feature extractor head" \Phi_{h,2} here? Please give clear details. These details are crucial for understanding the proposed method. 3. Equation (1) does not serve any practical purpose (as it is not directly optimized) and slows down the reader. It will be clearer to just introduce the cluster-centers matrix `C` and say k-means followed by Hungarian min-cost matching is used. 4. "Figure 4" is placed before Figure 3--please re-order. 5. Give a clear and concise description of CED curves, NME, NMI, F-measure, including their precise definition and ranges. It is not clear what is being measure in what units, and if a lower of higher value is better. What is "NME" (used in fig. 3 and 5)? I was unable to find its definition---is it the same as NMI (is a typo)? Without this context it is difficult to evaluate the experiments presented in this work. Typos: 1. L145: "This Section..." => "This section..." 2. Figure 2 uses "NMI", where as fig. 3 uses "NME". 3. L146: backbone

Relation to Prior Work: Yes, good discussion is provided. However, the following citation and comparison is missing: 1. Wiles et al. Self-supervised learning of a facial attribute embedding from video. BMVC 2018.

Reproducibility: No

Additional Feedback: 1. Unclear how certain landmarks are "absent" in visualisations---it likely corresponds to thresholding the heatmaps' confidence. Please give details. 2. Good to visualise the evolution of keypoints through the various iterations of pseudo-labelling. 3. Details of clustering of detectors and a visualisation of the cluster evolution would be very insightful. ---- Post rebuttal update: There is consensus amongst the reviewers that the proposed method does make some important contributions, e.g. invariance to large viewpoint changes, discovering number of keypoints, detecting absence of keypoints. However, the reliance on SuperPoint makes it not directly comparable to completely unsupervised methods. I had found the writing to be lacking, but given that the authors have promised to release the code and address *ALL* the points in this review, I am inclined to update my rating to "marginally above" given the technical merit.

[Author Response · NeurIPS 2020]

We **thank all Rs** for their comments and for recognizing the novelty of our approach. ●
**GC1: On SuperPoint initialization** This is part of our framework

| Method | F-measure |
|--------|-----------|
| SuperPoint | 0.42 |
| R2D2 | 0.33 |
| SIFT | 0.12 |
| SURF | 0.14 |
| FAST | 0.13 |

*smallNORB*

but we do not claim any methodological contribution for it. It has limitations (e.g. on low-textured surfaces) but also can be seen as an advantage given the tremendous progress on generic landmark detection. SuperPoint is trained in a self-supervised manner. Other works use pre-trained networks too (e.g. [31] uses fully supervised net from other domain, and [13] VGG for perceptual loss). We also provide an ablation study in L.197 where SuperPoint landmarks are replaced by a mixture of noisy g.t. landmarks and random noise which show (see paper's Fig. 2d) that, when the initial keypoints have F-measure of $\sim 0.4$ (on AFLW) our method works well. As the **Table herein** shows, SuperPoint meets this requirement, R2D2 is close to this, but SIFT/SURF/FAST (ran over rebuttal period) provide too poor initialization.

● **GC2: Performance on MAFL/AFLW** It is true that on "easy" frontal facial datasets our method does not surpass previous methods and this is to be expected as prior works have made tremendous progress on such datasets. Our method outperforms SOTA by large margin on the more difficult large pose (with 3D rotations) LS3D and Human3.6M.

● **R1**● **R1.1**: *Low texture surfaces*: Indeed, the method will not work well where the generic detector fails. Object landmark detectors mostly aim to detect salient points (e.g. mouth/eye corners, knee joints). Low texture points are difficult even for the supervised case. We will discuss this. ● **R1.2**: *Why it works for 3D rots*: You're right, actually, it is because of all the points you mention. ● **R1.3**: *On MAFL/AFLW*: Please see **GC2**. Note, LS3D contains portion of 300W. ● **R1.4**: *On poor accuracy for face contour (LS3D Forward)*: You are right, we will provide the per-landmark errors to make this clear. ● **R1.5**: *On labelled data for Superpoint*: Superpoint is trained in a self-supervised manner. So we still believe it qualifies for unsupervised. However, we will make this clear. The edge detector used in Pathak et al CVPR17 learns from manually annotated segmentation masks. ● **R1.7**: *Labels for MAFL*: You're right.

● **R2** ● **R2.1**: *Some results not being impressive.*: Please see **GC2**. ● **R2.2**: *Dependency on SuperPoint*: Please see **GC1**. ● **R2.3**: $K$ : $K$ is the underlying number of "discoverable" object landmarks which our method aims to discover, not a hyperparameter. In practice, this number mostly depends on the number of "good initial" landmarks detected by the generic detector. See also GC1. We will make this very clear, thank you. ● **R2.4**: *Non-differentiable process*: Clustering followed by the Hungarian algo. are performed at the end of each training round. They are not part of training the network; they just provide the pseudo-labels for the next iteration. We will make this clear, thank you.

● **R3**● **R3.1**: *SuperPoint:*: Please see **GC1**. ● **R3.2**: *only 1 3D dataset used:*: Our method outperforms SOTA by large margin on **2 datasets** with 3D rotations, namely **LS3D and Human3.6M**. ● **R3.3**: *Guarantee of semantic meaning*: We will rephrase to relax this statement. Our method however achieves a much stronger semantic representation, see L31-L33. ● **R3.4**: *Comparison with [22]*: The authors of [22] provide a non-complete Github repo (work-in-progress) at the time of submission. ● **R3.5**: *Schematic view*: We will try to fit one. ● **R3.6**: *Inconsistent language*: We will improve this. ● **R3.7**: *Readability (5, 6 and 11)*: Thank you we will clarify/improve ● **R3.8**: *Comparison with SIFT*: Please see **GC1** ● **R3.9**: *R2D2 yielding worse results*: We attribute this to the repeatability constraint of R2D2 which leads to sparse keypoints and poorer init. See also **GC1**. ● **R3.10**: *Background points for [13]*: We used their version of the code treating each frame separately for the sake of a fair comparison to all other methods. In this case [13] can yield points in background. We will clarify. ● **R3.11**: *Comparison on 3D databases*: Please see R3.2. We also conducted the experiment from [13] on smallNORB. Our method works well, some results are shown in **Figure herein**; compare to Fig. 6 from [13]. ● **R3.12**: *Comparison with [35]*: [35] represents a simpler setting than ours as it uses 3D objects rendered into 2D images assuming known 3D transformations between them. The objects are pre-segmented. We used significantly more complex real-world images (including human poses) without any knowledge of 3D information.

● **R4**● **R4.1**: *Additional complexity*: It is likely that the pipeline can be simplified. However, our method is the first of its kind to explore ideas like self-training, and correspondence via clustering to solve this problem. We show large improvements on difficult large pose datasets (LS3D, Human3.6M) so we believe that the impact of our results is significant. ● **R4.2**: *Performance on MAFL/AFLW*: Please see **GC2**.● **R4.3**: *Comparison with [31]*: [31] uses for initialization a net trained in a fully supervised manner that's why we didn't compare. Such an idea could fit our method too, but this is out of the scope of our paper. ● **R4.4**: *Resolution* : Methods are retrained using the provided codes for the same resolution (256px). We used $d = 12$px for resolution 256, equivalent to $d = 6$px for resolution 128 used in [13]. ● **R4.5**: *Clustering in 2nd phase*: No clustering is performed in 2nd phase. Clusters are just merged. A detector is learned via self-training with correspondences from 1st step. ● **R4.6**: *Comparison with SIFT*: Please see **GC1**. ● **R4.7**: *Clarity/reproducibility*: Thank you for such detailed feedback to improve our work! We will clarify ALL points raised and **we will release code**. ● **R4.8**: *NMI*: It's not typo, see suppl. L.18-L.29. ● **R4.9**: *Wiles et al.*: We will include it. ● **R4.10**: *Missing landmarks:*. Yes, we only visualize landmarks with confidence 0.2. ● **R4.11**: *Visualization of keypoints/cluster evolution:*. We will include a vis. of detected keypoints over iterative rounds. Vis. of cluster evolution is provided in Fig.2 (a). We will extend this to features learned over iterative rounds.

[Meta-Review · NeurIPS 2020]

This submission proposes an approach to unsupervised object landmark discovery. It initially received four reviews with mixed positive and negative scores (6,7,5,5). The rebuttal addressed some of the remaining concerns, which resulted in an increase in scores to (7,7,6,6). For these reasons, the AC's recommendation is to accept this submission for presentation as a poster, with a request for the authors to carefully revise the manuscript for the camera ready version to address the remaining concerns of the reviewers and improve the presentation clarity.